# TwinTrack: Post-hoc Multi-Rater Calibration for Medical Image Segmentation

**Tristan Kirscher**[1,2] (ID)                              TRISTAN.KIRSCHER@UNISTRA.FR
**Alexandra Ertl**[3,4,5] (ID)
**Klaus Maier-Hein**[3,4] (ID)
**Xavier Coubez**[2] (ID)
**Philippe Meyer**[1,2] (ID)
**Sylvain Faisan**[1] (ID)

[1] *ICube Laboratory, CNRS UMR-7357, University of Strasbourg, Strasbourg, France*

[2] *CLCC Institut Strauss, Strasbourg, France*

[3] *German Cancer Research Center (DKFZ) Heidelberg, Division of Medical Image Computing, Heidelberg, Germany*

[4] *Pattern Analysis and Learning Group, Department of Radiation Oncology, Heidelberg University Hospital, Heidelberg, Germany*

[5] *Medical Faculty Heidelberg, Heidelberg University, Heidelberg, Germany*

**Editors:** Accepted for publication at MIDL 2026

## Abstract

Pancreatic ductal adenocarcinoma (PDAC) segmentation on contrast-enhanced CT is inherently ambiguous: inter-rater disagreement among experts reflects genuine uncertainty rather than annotation noise. Standard deep learning approaches assume a single ground truth, producing probabilistic outputs that can be poorly calibrated and difficult to interpret under such ambiguity. We present TwinTrack, a framework that addresses this gap through post-hoc calibration of ensemble segmentation probabilities to the empirical mean human response (MHR) – the fraction of expert annotators labeling a voxel as tumor. Calibrated probabilities are thus directly interpretable as the expected proportion of annotators assigning the tumor label, explicitly modeling inter-rater disagreement. The proposed post-hoc calibration procedure is simple and requires only a small multi-rater calibration set. It consistently improves calibration metrics over standard approaches when evaluated on the MICCAI 2025 CURVAS–PDACVI multi-rater benchmark.

**Keywords:** PDAC, CT segmentation, uncertainty, calibration

## 1. Introduction

Pancreatic ductal adenocarcinoma (PDAC) is among the most lethal malignancies, and delineation of tumor extent on contrast-enhanced CT is important for staging and treatment planning. Yet, PDAC boundaries are often ill-defined, producing substantial inter-rater disagreement even among experts (Reuzel et al., 2021). In this setting, disagreement should not be treated purely as label noise: it also reflects genuine ambiguity in the image.

Most segmentation models are trained against a single target mask, and their output scores are often interpreted as confidence in a unique ground-truth boundary. For ambiguous tasks, this interpretation is unsatisfactory because poorly calibrated scores can overstate certainty (Guo et al., 2017; Ji et al., 2021). Prior work has mainly addressed ambiguity

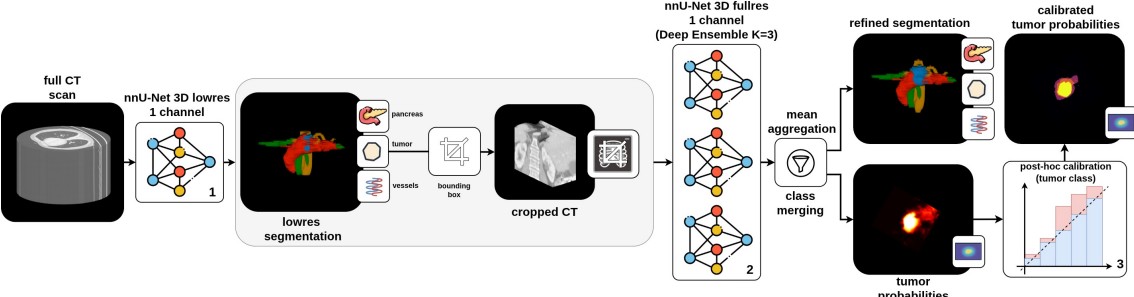

Figure 1: **TwinTrack pipeline.** A coarse model defines a high-recall ROI (1), followed by a high-resolution ensemble (2), with post-hoc PDAC calibration to the MHR (3).

during training, for example by learning from multiple annotations or modeling diverse plausible segmentations (Jensen et al., 2019; Ji et al., 2021; Kohl et al., 2018; Wu et al., 2024). Post-hoc calibration methods have also been studied for medical image segmentation, but primarily in single-rater settings (Mehrtash et al., 2020; Rousseau et al., 2025). Here, we instead study a simple post-hoc multi-rater calibration strategy: calibrating a fixed segmentation model to the voxelwise mean human response (MHR) using a small calibration set. To our knowledge, this setting has not been explicitly formalized in prior work, and Appendix A justifies the MHR as the calibration target under a multi-rater objective.

## 2. Method

TwinTrack combines a conventional two-stage segmenter with a lightweight multi-rater-aware calibration layer (Fig. 1). A low-resolution nnU-Net (Isensee et al., 2021) first localizes the pancreas at whole-volume scale and defines a high-recall region of interest by dilation. Within this region, an ensemble of $K=3$ independently trained high-resolution nnU-Nets refines the prediction, and their outputs are averaged to produce a pooled voxelwise score (Lakshminarayanan et al., 2017). Finally, the class outputs are merged to obtain a binary segmentation, distinguishing only between tumor and non-tumor voxels, with $\hat{y}(x)$ denoting the probability of the tumor class at voxel $x$.

The central contribution is a post-hoc calibration step, which uses isotonic regression to align the tumor probability with the mean human response (MHR): $\bar{y}(x) = \frac{1}{N} \sum_{i=1}^{N} y_i(x)$, where $\{y_i(x)\}_{i=1}^{N}$ denote the binary annotations of $N$ raters (1=tumor, 0=background) at voxel $x$, i.e., $\bar{y}(x)$ is the fraction of experts labeling voxel $x$ as tumor. A key property of isotonic regression is that it is monotone and thus preserves the ranking of voxel-wise predictions, changing only their probabilistic interpretation.

Aligning predictions with the MHR has several advantages: **(i) Robustness to rater disagreement:** Voxels with high inter-rater variability are mapped to intermediate probabilities, reflecting uncertainty rather than forcing an arbitrary hard label. **(ii) Meaningful probabilistic interpretation:** The calibrated output can be directly interpreted as the expected fraction of raters who would label a voxel as tumor. **(iii) Grounding in multi-**

Table 1: **Main results on the CURVAS–PDACVI test set ($n = 64$).** All methods use the same coarse-to-fine segmenter and differ only in the post-hoc calibration target. Values are bootstrap means over 5000 resamples. $^\dagger$ indicates a significant degradation relative to TwinTrack under paired bootstrap analysis (95% confidence interval (CI) of the difference excluding 0). Best and worst displayed values per metric are highlighted. Full CIs are reported in Appendix C.

| Calibration target | TDSC ↑ | ECE ↓ | CRPS ↓ | PORTA ↓ | SMV ↓ | AORTA ↓ | CELIAC ↓ | SMA ↓ |
|---|---|---|---|---|---|---|---|---|
| None (uncalibrated) | $0.553^\dagger$ | $0.0156^\dagger$ | 6032 | 32.7 | 34.8 | 6.34 | $18.6^\dagger$ | 33.1 |
| Single-rater | $0.300^\dagger$ | $0.0209^\dagger$ | $10342^\dagger$ | 38.5 | $48.7^\dagger$ | 9.09 | 19.9 | 30.7 |
| Hard-label | $0.307^\dagger$ | $0.0209^\dagger$ | $9860^\dagger$ | 34.2 | 42.3 | 7.29 | 17.9 | **25.9** |
| **MHR (TwinTrack)** | **0.569** | **0.0147** | **5924** | **28.9** | **32.8** | **6.10** | **14.5** | 28.7 |

**rater optimization:** As shown in Appendix A, calibrating to the MHR follows directly from a multi-rater isotonic regression objective.

## 3. Experiments and Results

Segmentation models are trained on PANORAMA batch 4 (PANORAMA Consortium, 2024). Multi-rater annotations are used only for post-hoc calibration: the CURVAS–PDACVI (Riera-Marín et al., 2026) training split provides 5 expert annotations for 40 CT scans, and the calibration mapping is fitted on that split without retraining. All methods in Table 1 use the same coarse-to-fine segmentation pipeline and differ only in the calibration target. Besides TwinTrack, which calibrates to the voxelwise MHR, we consider two alternatives: a *single-rater target*, where a calibration model is trained using the binary annotation of a single rater, and a *hard-label target*, where separate calibration models are trained for each rater's binary annotation, and their outputs are averaged at inference time. Table 1 reports bootstrap mean performance over 5000 resamples on the CURVAS–PDACVI test set ($n = 64$), focusing on TDSC (Thresholding Dice Score) for soft multi-rater segmentation, ECE (Expected Calibration Error) and CRPS (Continuous Ranked Probability Score) for probabilistic calibration, and vessel-specific vascular invasion (VI) metrics. Metric details, full quantitative results with 95% confidence intervals, and qualitative examples are provided in Appendices B, C, and E.

TwinTrack, i.e. post-hoc calibration to the MHR, achieves the best overall performance across ambiguity-aware metrics. Relative to the uncalibrated pipeline, it improves TDSC, lowers ECE, and lowers CRPS. It also yields the best VI performance for four of the five vessels (PORTA, SMV, AORTA, and CELIAC), while the hard-label target performs best only for SMA. Although the absolute ECE reduction is modest, as expected in a background-dominated voxel distribution, it is nevertheless significant, and the reliability diagram in Appendix D shows a clear improvement. Alternative calibration targets perform substantially worse: calibrating to a single-rater target or to hard binary annotations markedly reduces TDSC and worsens both ECE and CRPS, indicating that the MHR is a more suitable calibration target in this ambiguous multi-rater setting. Consistent with these results, our approach achieved the top ranking in the MICCAI 2025 CURVAS–PDACVI challenge.

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

## Appendix A. Why calibrate to the mean human response (MHR)?

### A.1. Notation

In our multi-rater setting, each voxel $x$ has binary annotations $\{y_i(x)\}_{i=1}^{N}$, with $y_i(x) \in \{0, 1\}$ (1=tumor, 0=background), and a model prediction $\hat{y}(x) \in [0, 1]$ for the tumor class.

We denote by $V$ the set of all voxels in the calibration dataset, and $|V|$ its total number of voxels. Voxels are partitioned into $M$ equal-mass bins $\{B_b\}_{b=1}^{M}$, such that each bin $B_b \subset V$ contains approximately the same number of voxels sorted by predicted probability. The weight of each bin is defined as

$$w_b = \frac{|B_b|}{|V|},$$

i.e., the fraction of voxels in the dataset that fall into bin $b$. For each bin $B_b$, we define the average predicted confidence

$$\hat{c}_b = \frac{1}{|B_b|} \sum_{x \in B_b} \hat{y}(x),$$

and the empirical positive rate of rater $i$:

$$\text{acc}_{b,i} = \frac{1}{|B_b|} \sum_{x \in B_b} y_i(x).$$

### A.2. Calibration with a single rater

If we consider a single rater, isotonic regression aims to learn a monotone mapping $m : [0, 1] \to [0, 1]$ by minimizing

$$\sum_{b=1}^{M} w_b \left( m(\hat{c}_b) - \text{acc}_{b,i} \right)^2. \tag{1}$$

Without the monotonicity constraint, the problem admits the trivial solution $m(\hat{c}_b) = \text{acc}_{b,i}$. Enforcing monotonicity leads to a constrained least-squares problem, which can be efficiently solved using the Pool-Adjacent-Violators algorithm (PAVA) (Barlow et al., 1972).

PAVA operates on the ordered sequence of bins and estimates $m(\hat{c}_b)$ for each bin $b$, depending only on the weights $w_b$ and the empirical accuracies $\text{acc}_{b,i}$. This produces a piecewise constant solution defined on the bins. The predicted confidences $\hat{c}_b$ can be used to construct a continuous mapping $m$ from the pairs $\{(\hat{c}_b, m(\hat{c}_b))\}_{b=1}^{M}$ using linear interpolation, resulting in a continuous, piecewise linear approximation of the isotonic solution.

Importantly, the use of binning is not only a computational convenience but also acts as an implicit regularization. In dense prediction settings such as medical image segmentation, the number of voxels can be extremely large and highly correlated. Performing isotonic regression directly at the voxel level would be both computationally expensive and prone to overfitting. Aggregating voxels into equal-mass bins stabilizes the empirical estimates $\text{acc}_{b,i}$ and leads to more robust calibration.

### A.3. Calibration with several raters

A natural multi-rater extension consists in learning a monotone mapping that minimizes:

$$\sum_{b=1}^{M} \sum_{i=1}^{N} w_b \left( m(\hat{c}_b) - \text{acc}_{b,i} \right)^2. \tag{2}$$

Using a standard decomposition of the sum of squares, we obtain:

$$\sum_{i=1}^{N} w_b \left( m(\hat{c}_b) - \text{acc}_{b,i} \right)^2 = N w_b \left( m(\hat{c}_b) - \bar{y}_b \right)^2 + C_b, \tag{3}$$

where $C_b$ is a constant independent of $m$, and

$$\bar{y}_b = \frac{1}{N} \sum_{i=1}^{N} \text{acc}_{b,i}.$$

The quantity $\bar{y}_b$ corresponds to the average fraction of raters labeling voxels in bin $b$ as tumor, i.e., the average of the mean human response (MHR) over voxels in $B_b$ (see Sec. 2).

Consequently, up to a multiplicative constant, the optimization problem reduces to:

$$\sum_{b=1}^{M} w_b \left( m(\hat{c}_b) - \bar{y}_b \right)^2. \tag{4}$$

This shows that multi-rater calibration is equivalent to calibrating predictions toward the MHR. Hence, calibrating toward the MHR is theoretically justified and can be performed using the same isotonic regression procedure as in the single-rater case.

## Appendix B. Evaluation Metrics

We follow the official CURVAS–PDACVI evaluation protocol and report four complementary metrics; for exact formulas, implementation details, and edge-case handling, we refer to the challenge documentation. For completeness, the official challenge testing-phase leaderboard is available on Grand Challenge (CURVAS–PDACVI Challenge Organizers, 2025, 2026).

**TDSC (soft overlap).** Multi-rater Dice score obtained by thresholding the predicted probability map and the MHR at multiple operating points and averaging the resulting Dice values.

**ECE (calibration).** Expected calibration error (Guo et al., 2017) computed per rater and then averaged across raters (official challenge metric). Concretely, for each rater $i$, we partition voxel predictions into 50 uniform bins, compute $\sum_b w_b |\hat{c}_b - \text{acc}_{b,i}|$, and then average the resulting ECE values over raters.

**CRPS (volume uncertainty).** Continuous ranked probability score between the predicted soft tumor volume and the reference volume distribution fitted from multi-rater annotations (Riera-Marín et al., 2025).

**VI (vascular invasion).** Vascular invasion is evaluated independently for five vessels (PORTA, SMV, AORTA, CELIAC, SMA). For each vessel, invasion is defined as the most restrictive involvement across axial, sagittal, and coronal planes. Ground truth and prediction invasion distributions are compared using the Wasserstein distance.

## Appendix C. Quantitative Results

Table 2 reports the full quantitative results with 95% bootstrap confidence intervals (CIs).

Table 2: **Quantitative results on the CURVAS–PDACVI test set ($n = 64$).** All methods use the same coarse-to-fine segmenter and differ only in the post-hoc calibration target. Values are bootstrap means over 5000 resamples with 95% CIs. Best and worst displayed values per metric are highlighted.

| Calibration target | TDSC ↑ | ECE ↓ | CRPS ↓ | PORTA ↓ | SMV ↓ | AORTA ↓ | CELIAC ↓ | SMA ↓ |
|---|---|---|---|---|---|---|---|---|
| None (uncalibrated) | 0.553 [0.478, 0.625] | 0.0156 [0.0127, 0.0191] | 6032.2 [3800.1, 9338.8] | 32.663 [23.892, 42.094] | 34.812 [26.317, 43.881] | 6.338 [3.431, 9.725] | 18.636 [10.182, 28.816] | 33.137 [22.301, 45.076] |
| Single-rater | 0.300 [0.258, 0.341] | 0.0209 [0.0176, 0.0248] | 10341.8 [7706.1, 13844.3] | 38.511 [27.935, 49.723] | 48.705 [38.005, 59.714] | 9.085 [4.776, 13.922] | 19.859 [11.403, 28.924] | 30.709 [21.085, 41.058] |
| Hard-label | 0.307 [0.266, 0.349] | 0.0209 [0.0176, 0.0248] | 9859.8 [7169.3, 13240.0] | 34.185 [24.511, 44.598] | 42.299 [32.570, 52.258] | 7.292 [3.808, 11.338] | 17.923 [10.362, 26.136] | **25.925** [17.468, 35.119] |
| **MHR (TwinTrack)** | **0.569** [0.496, 0.638] | **0.0147** [0.0117, 0.0182] | **5924.4** [3697.2, 9107.6] | **28.942** [21.484, 37.074] | **32.834** [24.477, 41.975] | **6.102** [3.475, 9.144] | **14.501** [7.850, 22.897] | 28.672 [19.768, 38.333] |

## Appendix D. Additional Calibration Diagnostics

For calibration learning, we use histogram binning with $M = 250$ equal-mass bins, selected on the training split as a good trade-off between granularity and stability of the learned mapping $m$. For test-set reliability visualization and ECE reporting, we use 50 uniform-width bins on $[0, 1]$, following the official evaluation protocol (Appendix B). Figure 2 compares the resulting test-set reliability curves for the uncalibrated ensemble and TwinTrack. It shows a clear improvement in calibration, with the TwinTrack curve much closer to the perfect calibration diagonal. Surprisingly, the ECE values remain very similar (see Tab. 1 and 2). This apparent discrepancy arises because the ECE is dominated by the heavily populated bin near 0, which mostly contains easy background voxels. As a result, improvements in less populated regions of the confidence range—where calibration matters most—have limited impact on the overall ECE. Note also that the ECE is computed independently for each rater and then averaged across raters (Appendix B). In contrast, the reliability diagram in Fig. 2 naturally extends to the multi-rater setting by plotting the empirical fraction of tumor labels (MHR) against the predicted confidence, following the formulation introduced in Appendix A.

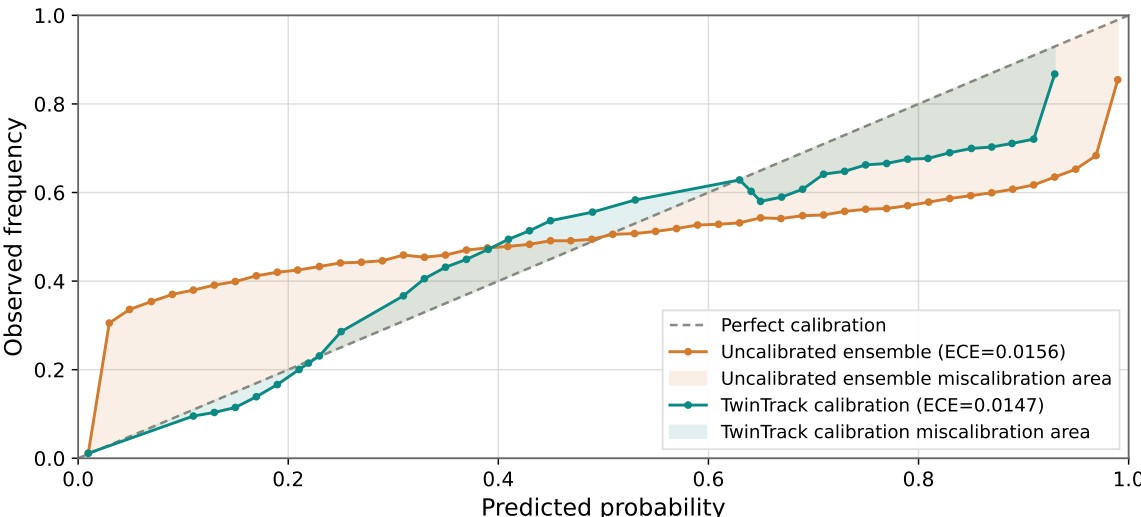

Figure 2: **Reliability comparison between the uncalibrated ensemble and Twin-Track calibration on the CURVAS–PDACVI test set.** The reliability diagram extends to the multi-rater setting by plotting the empirical fraction of tumor labels (MHR) against the predicted confidence, following the formulation introduced in Appendix A. TwinTrack calibration brings the curve closer to the perfect calibration diagonal, indicating improved alignment between predicted probabilities and expert consensus. Although the calibration mapping is monotone, residual non-monotonicity may persist in the reliability diagram due to finite-sample fluctuations, distribution shift, and differences in binning strategies between training (250 equal-mass bins) and evaluation (50 uniform-width bins).

## Appendix E. Qualitative Results

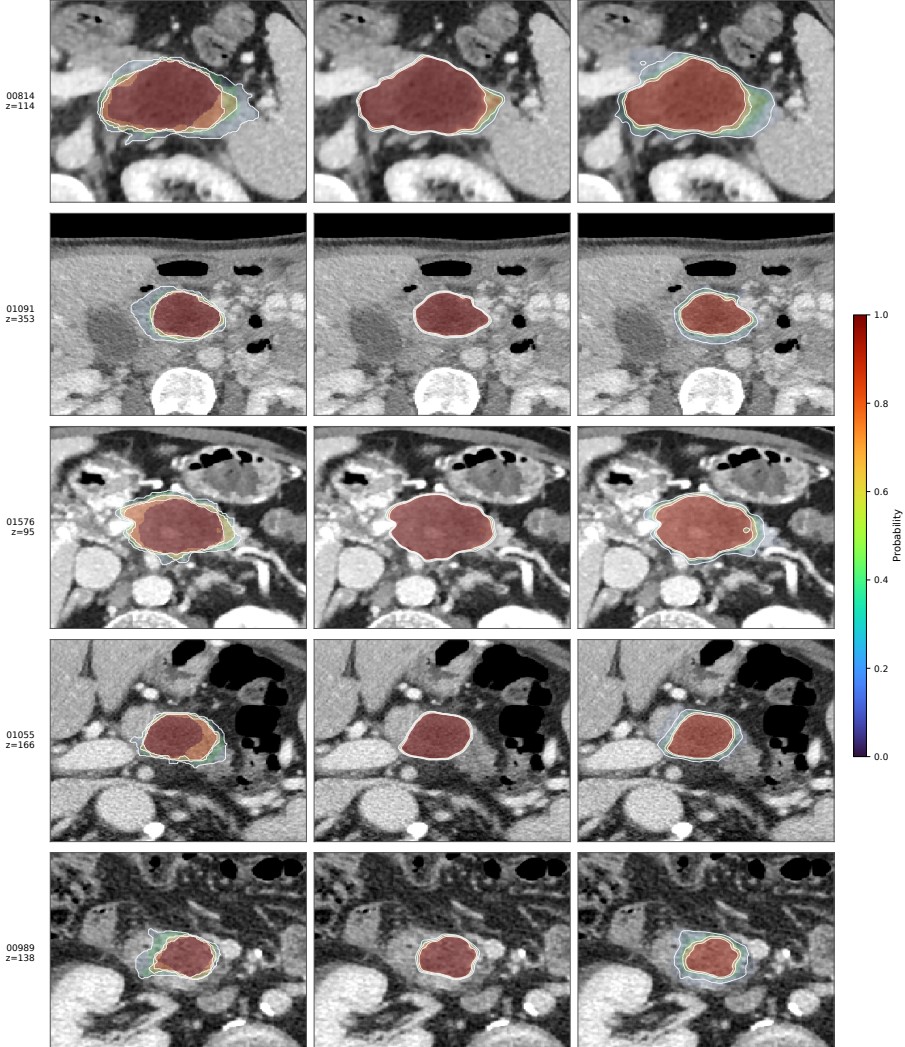

Figure 3: **Qualitative comparison on the CURVAS–PDACVI test set.** Each row shows one representative case (ID and axial slice index $z$ on the left), with all panels displaying the same zoomed lesion region. Left: mean human response (MHR) on CT, computed as the voxel-wise mean of the 5 expert annotations. Middle: uncalibrated TwinTrack confidence on CT. Right: calibrated TwinTrack confidence on CT. The uncalibrated model is visually overconfident relative to the MHR, whereas the calibrated predictions are more consistent with the graded uncertainty and spatial extent of expert disagreement. Overlay colors encode probability values in $[0, 1]$, with white contours indicating iso-probability levels.

