# OpenReview forum: "TwinTrack: Post-hoc Multi-Rater Calibration for Medical Image Segmentation"
_MIDL.io/2026/Short_Papers — MIDL 2026 - Short Papers Poster_

### Official Review · Reviewer_hqyz · 2026-05-03
**Potentially interesting but difficult to follow and effectively over the page limit**

**Rating:** 1
**Confidence:** 4

**Review:**

See strenghts and weaknesses. Overall, the paper could have been (re)structured to focus more on the important points.

**Summary:**

This paper addresses the topic of calibration of tumor segmentation, when there is potential rater disagreement on the exact contour of the objects to segment. The authors propose a post-hoc calibration technique, relying on a two-stages segmentation pipeline.


A lot of the content of this paper is in Appendix, to the point where I do not think it is self-contained anymore.

**Strengths:**

- The topic is very relevant, addressing one of the main roadblock toward widespread clinical adoption of quantification models into the clinical practice.
- The method had apparently good ranking in a public challenge

**Weaknesses:**

- The inter-rater agreement/disagreement of this specific dataset is not mentioned, and difficult for the reader to estimate
- The improvements over the uncalibrated network are fairly small (surprisingly), see Table 1
- The methodology remains unclear

**Justification Of Rating:**

The paper is effectively over the page limit ; key results are put in the Appendix (when it should be self-contained). Moreover, when looking at Table 1, the results are fairly marginal compared to the uncalibrated baseline ; which questions the method (itself not explained in enough details to be understood here). A better written paper could have clarified this, but as it stands, it is not a clear nor sufficient write-up about the work presented.

Per the very clear [author instructions](https://2026.midl.io/author-instructions):
> Short papers are strictly limited to 3 pages (excluding references). You may include appendices, but note that the main manuscript must be self-contained and the reviewers are not obliged to check the appendix.

---

### Decision · Program_Chairs · 2026-05-08

Accept (Poster)